# Disaster Risk Regionalization and Prediction of Corn Thrips Combined with Cloud Model: A Case Study of Shandong Province, China

Yanan Zuo [1], Fengxiang Jin [1], Min Ji [1,*], Zhenjin Li [1] and Jiutao Yang [2]

1   College of Geodesy and Geomatics, Shandong University of Science and Technology, Qingdao 266590, China; yananzuowork@163.com (Y.Z.)
2   Shandong Agricultural Technology Extension Center, Jinan 250000, China
*   Correspondence: jamesjimin@126.com

**Abstract:** Corn thrips do serious harm to the yield and quality of corn. In this paper, the Shandong Province of China was taken as the study area. Based on the data of the occurrence of corn thrips in Shandong Province, a risk regionalization model was established by using eight indicators under four categories of hazard, sensitivity, vulnerability and the disaster prevention and mitigation capacity of diseases and pests on a monthly time scale. Firstly, the cloud model was introduced to determine the weight of each indicator, and then the risk regionalization of the corn thrips disaster in Shandong Province was carried out using the weighted percentage method, the weighted comprehensive evaluation method and the natural disaster risk index method. Finally, combined with the collected data, the disaster prediction of corn thrip occurrence degree was realized based on multiple linear regression, genetic algorithm optimized back-propagation neural network and genetic algorithm optimized support vector machine methods. The results show that: (1) the risk of Corn thrips disaster is mainly concentrated in the central and western parts of Shandong Province. Heze City is a high-risk area. Liaocheng City, Dezhou City, Jinan City and Weifang City are relatively high-risk areas. (2) By comparing the prediction accuracy of the three models, it was determined that the genetic algorithm optimized support vector machine model has the best effect, with an average accuracy of 79.984%, which is 7.013% and 22.745% higher than that of the multiple linear regression and genetic algorithm optimized back-propagation neural network methods, respectively. The results of this study can provide a scientific basis for fine prevention of corn thrips in Shandong Province.

**Keywords:** corn thrips; risk regionalization; disaster risk prediction; cloud model; genetic algorithm optimized support vector machine

## 1. Introduction

With the increase in the global population, the demand for food is gradually increasing [1]. As one of the important food crops worldwide, corn plays an important role in food security [2]. Corn thrips (Thysanoptera, Thripidae) are one of the main pests in the corn seedling stage. The adult or nymphae thrips feed on seedling juice, so that the internode of the whole corn plant is shortened, swollen or curved, which seriously affects the yield and quality [3]. At present, corn thrips have caused serious harm in China, Europe, Africa and other regions [4–6]. Therefore, the effective assessment and prediction of the risk of corn thrips are of great significance for the protection of corn yield and food security.

Risk regionalization based on disaster data has been widely applied in many fields [7,8], especially in agriculture. Risk regionalization can provide effective data references for food security production. Zhang et al. [9] evaluated the current cropping systems and patterns in different agro-ecological zones in Sichuan Province by calculating the ensure index of precipitation, the yield variation coefficient, the ratio of sown area to cultivated area and the disaster resistance capability coefficient of agricultural planting areas in Sichuan Province.

Singh et al. [10] proposed the use of analysis and discussion methods of multi-scalar and multi-indicator evaluation. Through the evaluation and discussion of 26 indicators in four dimensions, the disaster resistance and recovery capability of the agricultural climate region in India was analyzed. Chen et al. [11] performed a detailed analysis of the stability and comparative advantage of rice production in 30 provinces on the basis of relative rice production data from 2000 to 2012 in China, adopted the non-parametric information diffusion model based on entropy theory to assess rice production risks, and used hierarchical cluster analysis to divide risk levels. Li et al. [12] applied fuzzy multiconnection theory and the analytic hierarchy process model (AHP) to evaluate the safety of the Chinese rapeseed supply chain from 2010 to 2020 through a comprehensive risk assessment model of the rapeseed supply chain, using two primary indicators and ten secondary indicators. Hudait et al. [13] evaluated the land suitability of areca leaf planting in the Tamluk subdivision, Purba Medinipur district, West Bengal, India, using 17 parameters of five categories. At present, many scholars have adopted the traditional AHP model [8,11–13]. Although this method is simple and practical, it is limited by the subjective cognition of decision makers, which often leads to bias in weight calculation and has a great influence on results [14].

The cloud model was proposed by Li Deyi [15], a member of the Chinese Academy of Engineering. Based on stochastic mathematics and fuzzy theory, the cloud model realizes the mutual conversion between qualitative and quantitative factors [16]. It is a model of uncertainty transformation between qualitative concepts and quantitative representations expressed in natural language values [15]. Aiming at the problem of weight calculation deviation, this method can effectively reduce the uncertainty of weight calculation and improve the accuracy of the weight calculation of the indicator. The cloud model has been applied well in ecological protection. Hu et al. [17] used uncertainty reasoning based on the cloud model to transform the conditions of uncertain factors into quantitative values and proposed a new method with which to solve the weight of land evaluation factors. Wang et al. [18] and Yao et al. [19] put forward the multidimensional normal cloud model (MNCM) and the multidimensional cloud model (MSCM), respectively, which have realized the eutrophication evaluation of lakes and reservoirs. In addition, the cloud model has been widely used in many fields such as transportation and industry [20,21]. Therefore, the cloud model is introduced into the disaster regionalization of corn thrips in this paper in order to achieve more reasonable regionalization results.

At present, the disease and thrips pest prediction models are relatively mature [22]. Many researchers have carried out intensive studies on the construction of disease and pest prediction model, and there is a systematic development trend [23]. Maiorano et al. [24] established a biological model of corn borer physiology based on the daily accumulated temperature in northern Italy. Cao et al. [25] built a diseases and pests prediction model based on the sparse clustering algorithm, which was better for this than the back-propagation (BP) neural network method. Gao et al. [26] used the observation data of corn borer infestation in Tonghua City of Jilin Province in China and related weather factors to establish a meteorological prediction model of the occurrence degree of corn borer prevalence by statistical means, and the accuracy of the model return test was 81.5%. Liu et al. [27] established a prediction model of forecasting powdery mildew and anthracnose based on long short-term memory technique with accuracies of 0.74 and 0.68, respectively. To sum up, influenced by the growth mechanism, the factors affecting the growth of different crops are quite different, which also leads to differences in model construction [28–30]. Similarly, for different diseases and pests, the disaster risk prediction accuracy of each model is inconsistent. At present, there is a lack of research on disaster prediction models of corn thrips, and a model which is more accurate and effective still needs to be developed.

The occurrence of pests is complex and uncertain. Multiple linear regression techniques can explore the correlation between the occurrence of corn thrips and its influencing factors based on relevant data and establish a prediction regression model. In this paper, a prediction model for the occurrence of corn thrips was constructed using multiple linear regression techniques to reflect corn thrip linear characteristics in the ecosystem

and explore the linear relationship. Yang et al. [31] used multiple linear regression to establish a predictive model for the occurrence of second-generation corn borers, achieving higher short-term prediction accuracy than previous similar predictions. This verified the feasibility of using multiple linear regression to predict the occurrence of pests. A BP neural network [32] is a multi-layer feedforward neural network that automatically updates weight and threshold values through the forward propagation of signals and backward propagation of errors. Its neuron activation function is a sigmoid function. It can approximate and simulate any nonlinear function of pest occurrence and other nonlinear dynamic phenomena without considering the internal structure of a given mathematical model, assuming premise conditions or artificially determining factor weights [33]. Considering the powerful self-organizing and self-learning abilities of BP neural network and its superiority in solving nonlinear problems, it is a very suitable method for constructing a nonlinear prediction model for the occurrence of corn thrips. Support vector machine (SVM) [34] is a machine learning method based on the principle of structural risk minimization. It maps data onto a high-dimensional feature space through a nonlinear mapping function and performs linear regression in this space. It does not require any assumptions about the distribution properties of data and solves the problems of small sample size, nonlinearity, overfitting, curse of dimensionality, and local optimum, which can be used for both linear and nonlinear data. Since the occurrence data of corn thrips in this paper have the typical small sample problem, the SVM method is selected to construct the prediction model. In summary, since the population dynamics of corn thrips and their influencing factors constitute a complex linear or nonlinear structure, this paper selects three methods with different principles, including multiple linear regression, genetic algorithm optimized back-propagation (GA-BP) neural network and genetic algorithm-optimized support vector machine (GA-SVM), to construct a prediction model for the occurrence of corn thrips.

Shandong Province is the main growing and producing area of corn in China, and corn thrips cause different degrees of loss every year. It is of great significance to evaluate and predict the risk of corn thrip occurrence in Shandong Province effectively. Therefore, in order to accurately reflect the occurrence of corn thrips each month, this paper took Shandong Province as the study area and constructed risk regionalization and prediction models of corn thrips in Shandong Province, seeking to provide a scientific basis for the prevention and control of corn thrips in Shandong Province.

The structure of this paper is as follows: the second chapter introduces the study area and data source. The third chapter outlines the study methods, including data preprocessing, disaster risk regionalization methods and disaster risk prediction models. The fourth chapter overviews the study results, including the weight distribution value of each indicator after cloud model processing, the results of disaster risk regionalization of corn thrips in Shandong Province and the prediction results. These are combined with the true values to conduct quantitative analysis of different prediction models, screening for the optimal model. Finally, some valuable conclusions are given in this paper.

## 2. Study Area and Data

### 2.1. Study Area

Shandong Province is located on the east coast of China and the lower reaches of the Yellow River, between 34°25′ N and 38°23′ N, 114°36′ E and 112°43′ E. The longest section of the province runs 700 km from east to west, and the widest runs 420 km from north to south, with a total land area of $15.67 \times 10^4$ km$^2$ and abundant natural resources [35]. Shandong is a major province of corn production and consumption, with a perennial planting area of about $2.7 \times 10^7$ hm$^2$ [36]. Shandong possesses a temperate monsoon climate, with concentrated precipitation, rain and heat in the same season. The average annual temperature is 11 °C~14 °C, the average annual precipitation is generally between 550 and 950 mm, which concentrated in summer, and the annual light duration is 2290~2890 h. The natural conditions are very suitable for the growth of corn. The adult peak period of corn thrips is in June and July, when the summer corn seedling stage and spring corn core

leaf stage seriously harm the quality and yield of corn. Therefore, this paper focuses on exploring the disasters caused by corn thrips in June and July.

### 2.2. Species Profile

Thrips are pests of corn seedlings, and include corn yellow thrips, grass thrips, and rice stem thrips. They are small in size and can fly and jump well. The female adults are divided into long-winged, half-long-winged, and short-winged types. They have small bodies, a dark yellow color, and dark gray spots on the chest. The forewings are grayish-yellow, long and narrow, with few but distinct wing veins and long marginal hairs. The half-long-winged type has wings that only reach the 5th segment of the abdomen, while the short-winged type has wings that are slightly triangular and bud-like in shape. Figure 1a is an enlarged photo of a corn thrips taken in the study.

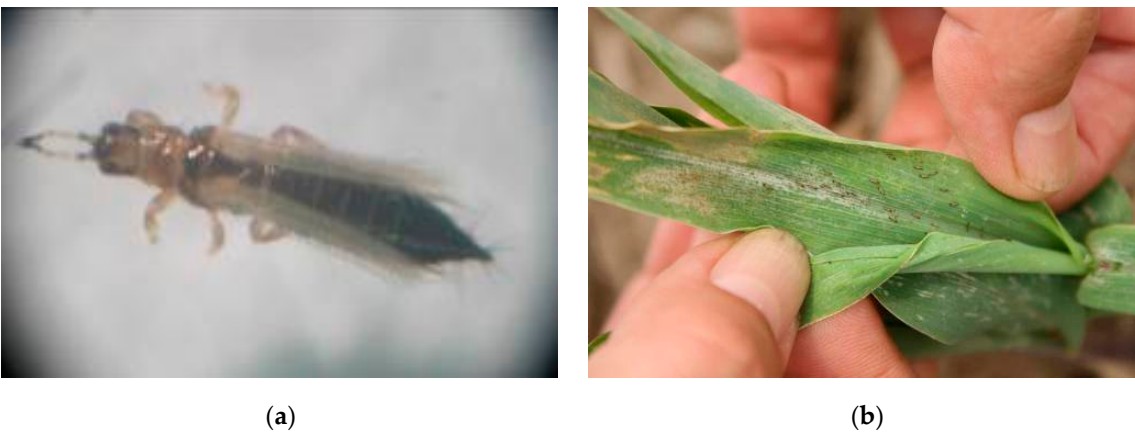

(**a**)　　　　　　　　　　　　　　　　　　　　　　　　　　(**b**)

**Figure 1.** Field investigation photos. (**a**) Photo of corn thrips; (**b**) Photo of corn affected by corn thrips.

When damaging corn, corn yellow thrips first harm the back of the leaves; grass thrips and rice stem thrips first harm the front of the leaves. Both can cause intermittent silver-white streaks on the leaves, accompanied by small spots. Corn thrips are mostly concentrated on corn tender leaves and heart leaves. After corn leaves are damaged, they show pale stripes. In severe cases, the half-part of the leaf end dries up, and the heart leaves cannot be pulled out. Even if pulled out, thrips often cause a large number of dead seedlings in a bushy top-like manner, resulting in the destruction of the crops. In addition to harming corn, corn thrips also damage various grass plants such as wheat, rice, and millet. Some studies have shown that the feeding damage caused by corn thrips can lead to a decrease in corn yield by about 10–20% [37,38], causing very serious practical harm. Figure 1b shows corn damaged by thrips.

### 2.3. Data

The data used in this paper included diseases and pests occurrence data, corn means of production, socio-economic statistical data and monthly meteorological data. The data were all from 2013 to 2019. The disease and pest occurrence data were obtained from the "Plant Protection Situation in the Province" published by the Shandong Provincial Department of Agriculture and Rural Affairs. These included infested plant rate, strain rate, number of insects per 100 plants, and number of insects per plant. Corn production data and socio-economic statistical data were obtained from the "Shandong Statistical Yearbook", including corn yield, corn sown area and GDP per capita. The meteorological data comes from China Weather Data Network (http://data.cma.cn/) (18 February 2022), including 11 meteorological factors including extreme wind speed, maximum temperature, precipitation at 20–20 h, average air pressure, average temperature, average vapor pressure, average relative humidity, average maximum temperature, sunshine duration, maximum wind speed and maximum daily precipitation.

According to the "Grading Standard of the Occurrence Degree of Crop Diseases and Pests in Shandong Province" (DB37/T232-1996), the occurrence degree of corn thrips in Shandong Province was divided into five levels, namely light occurrence (Level 1), relatively light occurrence (Level 2), moderate occurrence (Level 3), severe occurrence (Level 4) and major occurrence (Level 5), as shown in Table 1.

**Table 1.** Grading table for the degree of corn thrips occurrence.

| Occurrence Indicator | Level of Occurrence Degree | | | | |
|---|---|---|---|---|---|
| | Level 1 | Level 2 | Level 3 | Level 4 | Level 5 |
| Infested plants rate (X,%) | $10 \leq X \leq 20$ | $21 \leq X \leq 40$ | $41 \leq X \leq 60$ | $61 \leq X \leq 80$ | $X > 80$ |

The data of "Plant Protection Situation in the Province" issued by the Shandong Provincial Department of Agriculture and Rural Affairs were sorted through, and the occurrence degree was graded according to Table 1. In this paper, the maximum occurrence degree per month in each city was taken as the time scale used to count the frequency of occurrence degree. After the grading work was completed, the frequency of the highest occurrence degree of corn thrips in June and July from 2013 to 2019 in each city was calculated as light occurrence (V1: Level 0 to 1 is light occurrence), moderate occurrence (V2: Level 2 to 3 is moderate occurrence), and severe occurrence (V3: Level 4 to 5 is severe occurrence), respectively. The obtained frequency of corn thrips is shown in Figure 2.

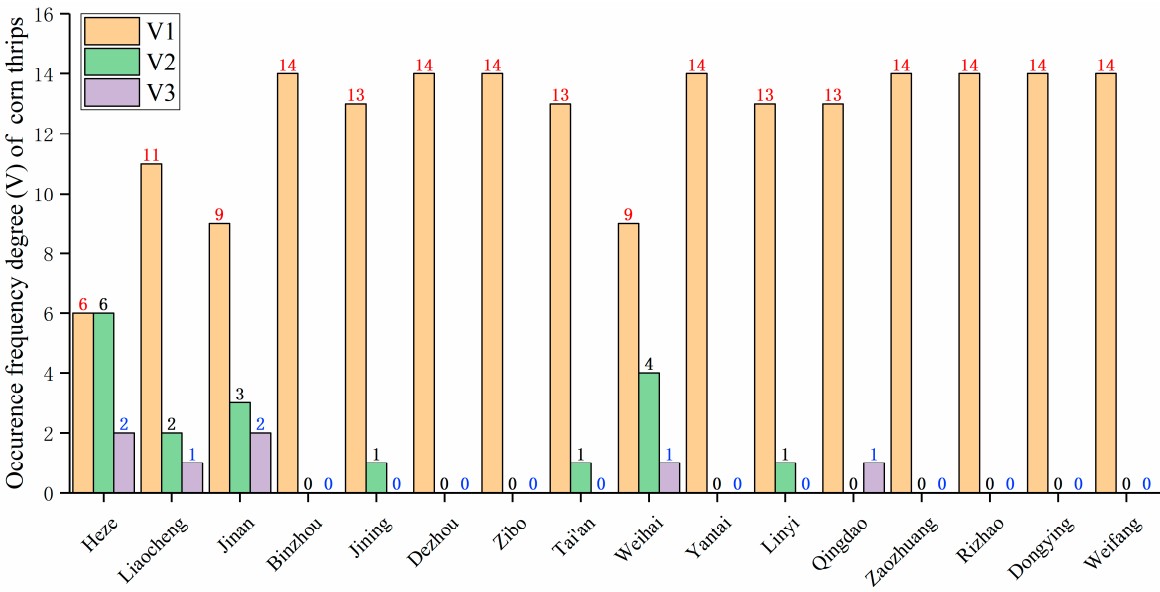

**Figure 2.** Occurrence frequency of corn thrips from 2013 to 2019. V1 represents light occurrence; V2 represents moderate occurrence; V3 indicates severe occurrence.

## 3. Methods

### 3.1. Data Normalization

Due to the difference in dimension between the indicators that affect the risk of corn thrips disaster, the normalization of the data is required in order to enable calculation and comparison across different data layers. This paper uses extreme difference normalization to convert the raster unit data to values in the range of [0,1]. Assuming that there are m evaluation indicators in the study area and n evaluation units, there is an original evaluation matrix:

$$R = (X_{ij})_{m \times n} (i = 1, 2, \ldots, m; j = 1, 2, \ldots, n) \tag{1}$$

where $X_{ij}$ is the raw data corresponding to the *j*-th unit of the *i*-th indicator. The following conversions are made using the extreme difference normalization method:

$$X_{ij}' = \frac{X_{ij} - \min\{X_{ij}\}}{\max\{X_{ij}\} - \min\{X_{ij}\}} \tag{2}$$

where $\max\{X_{ij}\}$ and $\min\{X_{ij}\}$ are the maximum and minimum values of all unit data under the *i*-th indicator, respectively; $X_{ij}'$ is the normalized unit data.

### *3.2. Disaster Risk Regionalization Methods*

### 3.2.1. Cloud Model

Although the traditional analytic hierarchy process (AHP) is simple and practical, it is limited by the subjective cognition of decision makers and has uncertainties, which often biases the weight calculation and has a significant impact on the results. The cloud model introduced in this paper can effectively reduce the uncertainty of weight calculation and improve the accuracy of weight calculation of risk indicators.

A cloud model has three digital features: Ex, En and He, through which the fuzziness and randomness of the described concepts can be reflected. It is a type of model that transforms a qualitative concept, expressed by language value and its quantitative representation. By combining the qualitative concept with the quantitative representation, the mapping relationship between the qualitative and quantitative is performed. The basic steps of the cloud model constructed in this paper are as follows:

1.  A hierarchical structure is constructed according to the natural characteristics of diseases and pests and the typicality of the study area, and the complex problem can be summed up as the total ranking problem of the weight order of the criterion layer and the indicator layer relative to the target layer.

2.  The cloud model is used to represent the language value. Each language variable corresponds to a cloud. At length, the golden section method is used to calculate the cloud evaluation scale. In this paper, according to the evaluation of the expert group, the expected value corresponding to the interval [0,10] was figured as the theoretical domain $[X_{min}, X_{max}]$, and the superentropy was determined as $H_{e0} = 0.05$. The evaluation scale were developed according to the operation rules shown in Table 2. The cloud evaluation scales were calculated as follows: $E_2$ is very important (10, 1.03, 0.13), $E_1$ is important (6.91, 0.64, 0.08), $E_0$ is relatively important (5, 0.39, 0.05), $E_{-1}$ is general (3.09, 0.64, 0.08), and $E_{-2}$ is unimportant (0,1.03, 0.13).

**Table 2.** Golden Section method of calculating cloud evaluation scales.

| Cloud | Expected Value ($E_x$) | Entropy ($E_n$) | Hyperentropy ($H_e$) |
|---|---|---|---|
| $E_2$ ($E_{x2}$, $E_{n2}$, $H_{e2}$) | $X_{max}$ | $E_{n1}/0.618$ | $H_{e1}/0.618$ |
| $E_1$ ($E_{x1}$, $E_{n1}$, $H_{e1}$) | $E_{x0} + 0.382 (X_{min} + X_{max})/2$ | $0.382 (X_{max} - X_{min})/6$ | $H_{e0}/0.618$ |
| $E_0$ ($E_{x0}$, $E_{n0}$, $H_{e0}$) | $(X_{min} + X_{max})/2$ | $0.618E_{n1}$ | $H_{e0}$ |
| $E_{-1}$ ($E_{x-1}$, $E_{n-1}$, $H_{e-1}$) | $Ex_0 - 0.382 (X_{min} + X_{max})/2$ | $0.382 (X_{max} - X_{min})/6$ | $H_{e0}/0.618$ |
| $E_{-2}$ ($E_{x-2}$, $E_{n-2}$, $H_{e-2}$) | $X_{min}$ | $E_{n1}/0.618$ | $H_{e1}/0.618$ |

3.  In the normal cloud model, the number of cloud drops *n* is first set to 1. Then, the model outputs the random quantitative value of the evaluation cloud when the number of cloud drops is 1, as is the degree of certainty corresponding to the quantitative value, on the basis of which the relative importance of the risk evaluation indicators is then calculated. Subsequently, when calculating the weights of the evaluation indicators, the number of cloud drops is entered and *n* = 10 is taken to convert the qualitative evaluation into a quantitative value.

4.  In the forward cloud model, the three eigenvalues corresponding to the indicator evaluation scale ($E_x$, $E_n$ and $H_e$) are input successively, and then the model outputs a point ($x$, $\mu$) determined in the cloud image. The number of evaluation indicators should be *m*, and the number of experts who separately evaluate the importance of indicators should be set as *u*. Thus, the number of natural language evaluation that

can be obtained from experts for a single evaluation indicator is $u$, that is, the number of cloud drops. Each evaluation indicator is transformed into $(x_i, \mu_i)$ by a cloud model.

5. The relative weight of each expert evaluation is calculated as:

$$\omega_i = \frac{\mu_i}{\sum\limits_{i=1}^{m} \mu_i} \tag{3}$$

6. The importance of evaluation indicator is calculated as:

$$c_{\omega_i} = \sum\limits_{i=1}^{m} \omega_i \, x_i \tag{4}$$

7. The relative importance of evaluation indicator, namely the weight of evaluation indicator, is calculated as:

$$C_{W_i} = \frac{c_{\omega_i}}{\sum\limits_{i=1}^{m} c_{\omega_i}} \tag{5}$$

### 3.2.2. Weighted Percentage

Using the damage data of corn thrips in 16 cities in Shandong Province from 2013 to 2019, based on the frequency (Figure 2) of different occurrence degrees of corn thrips over the years and the weight determined by the cloud model, the weighted percentage value was used to obtain the impact degree of corn thrips. This is expressed in the hazard index, also known as the impact index of diseases and pests. When the yield loss of severe occurrence is large, the weight coefficient will be large. When the yield loss of light occurrence is small, the weight coefficient will be small. The hazard index ($H$) of disaster-causing factors is calculated using the following equation:

$$H_j = \sum\limits_{i=1}^{3} \left\{ \frac{V_{ij}}{\sum V_i} \cdot P_i \right\} \tag{6}$$

where $H_j$ refers to the hazard index of corn thrips in city $j$; $V_{ij}$ refers to the frequency of occurrence degree $i$ in city $j$; $P_i$ refers to the weight of different frequency.

### 3.2.3. Weighted Comprehensive Evaluation

To obtain the weight coefficient of each indicator determined by the cloud model after processing, it should be multiplied with the quantitative value of each indicator of the corresponding evaluated object and then added to it. This method is called the weighted comprehensive evaluation method, and the equation for it can be expressed as:

$$C = \sum\limits_{i=1}^{n} (W_i \times P_i) \tag{7}$$

where $C$ is the evaluation indicator, $W_i$ is the normalized value of the $i$-th indicator, and $P_i$ is the weight of the $i$-th indicator factor. In this paper, the sensitivity of disaster-forming environments index was established by using the weighted comprehensive evaluation method.

### 3.2.4. Natural Disaster Risk Index

Natural disaster risk expresses the result of the comprehensive action of hazard, sensitivity, vulnerability and disaster prevention and mitigation capacity. The natural disaster index is usually used to represent the risk degree. It should be noted that the risk of corn thrips is reduced in areas with strong disaster prevention and mitigation capacity. The equation of the natural disaster risk index is:

$$R = H \times a + E \times b + V \times c + (1 - D) \times d \tag{8}$$

where $R$ is the comprehensive risk index of corn thrips; $H$ is the danger of the disaster-causing factors index; $E$ is the sensitivity of the disaster-forming environments index; $V$ is

the vulnerability of the disaster-bearing objects index; $D$ is the disaster prevention and mitigation capacity index; $a$, $b$, $c$ and $d$ are the weights of the corresponding indices. In this paper, the comprehensive risk assessment model of corn thrips was established using the natural disaster risk index method.

### 3.3. Disaster Risk Prediction Methods

In this paper, three prediction models were used to predict the occurrence degree of corn thrips. Through comparative analysis, the optimal precision model was selected for practical prediction in order to minimize the loss caused by corn thrips.

### 3.3.1. Multiple Linear Regression Model

In regression analysis, if there are two or more independent variables, the method is termed multiple regression. According to the biological characteristics of corn thrips, 11 meteorological factors during June and July of 2013 to 2017 were selected as input variables of the model. The average value of each meteorological factor in each month of the year was calculated. The maximum occurrence degree of each month in each city was calculated and the average value was calculated by taking Shandong Province as the unit. The correlation analysis of binary distance variables was used to obtain the correlation degree of 11 meteorological factors and the occurrence degree of corn thrips in June and July. With the occurrence degree of corn thrips as the dependent variable (y) and the meteorological factor with the greatest influence on the dependent variable of each month as the independent variable (x), multiple linear regression analysis was used to establish the disaster risk prediction model, as shown in Table 3.

**Table 3.** Correlation and prediction models for the effect of meteorological factors on the occurrence degree of corn thrips by month.

| Factor Name | June | July |
|---|---|---|
| Extreme wind speed ($x_1$) | −0.664 | −0.100 |
| Maximum temperature ($x_2$) | −0.188 | −0.348 |
| Precipitation at 20–20 h ($x_3$) | −0.365 | 0.675 |
| Average air pressure ($x_4$) | −0.106 | −0.676 |
| Average temperature ($x_5$) | 0.182 | −0.311 |
| Average vapor pressure ($x_6$) | 0.671 | −0.155 |
| Average relative humidity ($x_7$) | 0.327 | 0.175 |
| Average maximum temperature ($x_8$) | −0.071 | −0.255 |
| Sunshine duration ($x_9$) | 0.587 | 0.076 |
| Maximum wind speed ($x_{10}$) | −0.635 | 0.107 |
| Maximum daily precipitation ($x_{11}$) | −0.140 | 0.770 |
| Prediction equation | $y = -0.397 + 0.211x_6 - 0.097x_1$ | $y = -333.210 + 0.024x_{11} + 0.334x_4 + 0.004x_3$ |

### 3.3.2. GA-BP

Back-propagation (BP) neural networks [32] are mathematical models of nonlinear uncertainty. They are multi-layer feedforward neural network trained according to the error back-propagation algorithm, which is one of the most widely used neural network models at present. It consists of an input layer, hidden layer and output layer, with forward propagation and back-propagation included in the learning process.

Genetic algorithms (GA) constitute another artificial intelligence method used to simulate the biological evolution process. This method has the feature of global optimization. In this vein, it can overcome the defects of local optimization of BP algorithm well, optimize the initial weight and threshold of the BP neural network, improve the stability of BP neural network and shorten the time. Their basic principle involved the use of a genetic algorithm to optimize the weight and threshold of the BP neural network. When coding, the individual is the weight and threshold of the neural network. The training error of the

BP neural network is taken as the fitness value of the individual, and the optimal initial weight and threshold of the BP neural network are found through selection, crossover and variation operations.

### 3.3.3. GA-SVM

Support vector machines (SVM) [34] constitute a machine learning method based on the principle of structural risk minimization. They can solve problems such as small sample size, nonlinear, overfitting, dimension disaster and local optimum well, and have excellent generalization ability. Their basic principle involves the mapping of the data to a high-dimensional feature space through a nonlinear mapping function and the performance of linear regression in this space.

The modeling idea of genetic algorithm optimized support vector machine (GA-SVM) is as follows:

- The sample is divided into two parts: training set and test set.
- Determination of kernel function and parameters.
- Genetic algorithm to optimize model parameters.
- The SVM model is built using the optimized parameters, and the training samples are input to the model for learning.
- The optimal model is used to predict the test set.
- Model performance is evaluated.

### *3.4. Experimental Settings and Evaluation Measures*

#### 3.4.1. Training Platform and Hardware

The training platform of the experiment is a Windows 11 64-bit system, and the processor is AMD Ryzen 7 5800H (32G memory). The Python 3.6 platform was used to build the cloud model and the prediction models.

#### 3.4.2. Dataset

The three models in this study all used the occurrence degree of corn thrips in Shandong Province during June and July from 2013 to 2019 as the dependent variable, and the related meteorological factors affecting occurrence as the independent variables, to construct a sample set. The samples were divided into a training set and a testing set, with data from 2013 to 2017 used as the training set to establish the model and to predict the occurrence degree of corn thrips for 2018–2019, thus testing the prediction performance of the model.

#### 3.4.3. Model Hyperparameters Setting

In the process of building a multiple linear regression model, the step method selects the F-test, the F-to-enter value is set to 0.15, the F-to-remove value is set to 0.2, the independent variables are screened by the stepwise selection method, and the sig value of significance judgment is set to 0.05.

When building a GA-BP model, the greater the number of hidden layers included, the stronger the nonlinear mapping ability, although following this rubric means the network performance will be reduced. An excessively low number of hidden layer nodes would lead to excessively long training times. The number of neurons in the input layer of the BP neural network prediction model was determined by the meteorological factors that mainly affect corn thrips. Therefore, 11 input nodes were selected for the input layer. The occurrence degree of corn thrips was selected for the output layer, and the number of nodes in the output layer was set as 1. After running the program several times, the optimal number of hidden layer nodes was determined to be 5. Finally, in this study, a GA-BP neural network model was used to predict the corn thrips with a network structure of (11, 5, 1). The activation function of the hidden layer was determined to be tansig, while the activation function of the output layer was purelin. The maximum number of training iterations was set to 1000, with a learning rate of 0.1 and a training error performance of

$1 \times 10^{-20}$. The GA algorithm population size was 30, with 50 generations of evolution. The crossover probability is 0.4, and the mutation probability is 0.2.

When establishing the GA-SVM model, the first step is to screen the kernel function. Generally, the SVM model performance using the radial basis kernel function is superior to other kernel functions. Therefore, this paper selects the radial basis kernel function as the kernel function for the GA-SVM model. According to the SVM regression principle, it is known that the disaster risk prediction performance of the SVM model using the radial basis kernel function is closely related to the values of c (which mainly controls the degree of punishment for misclassified samples) and g (the parameter of the radial basis kernel function). To obtain the SVM model with an optimal performance, it is necessary to first obtain the optimal values of c and g. When optimizing SVM parameters using GA, the parameters c and g are treated as individual values in GA. The range of variation for parameter c is set to (0,100], and the range of variation for parameter g is set to [0,1000]. The mean square error is used as the individual fitness value and, after multiple generations of genetic evolution through selection, crossover, mutation, and other genetic operations, the most optimal parameter of the GA-SVM model is obtained. In this paper, the optimal parameters c = 70.0177 and g = 1.5221. For other parameters, the maximum evolution generation was set to 100, population size to 20, and generation gap to 0.9.

### 3.4.4. Evaluation Index

The test set was used to evaluate the performance of the prediction model in this study, and the prediction accuracy of the test set was used as the evaluation index of the model. The prediction accuracy equation is as follows:

$$predict\_accuracy = \left(1 - \left|\frac{y_i - \overline{y_i}}{y_i}\right|\right) \times 100\% \tag{9}$$

where, $y_i$ is the true value of occurrence degree and the $\overline{y_i}$ is the prediction value of the model. The higher the precision value, the better the prediction effect.

## 4. Results and Discussion

### 4.1. Indicator System Construction

The formation of corn thrips disaster is affected by many factors, which can be divided into dual categories of natural factors and human activities. In previous studies on crop diseases and pests risk regionalization, most of the risk assessment indicator systems were established from three aspects: disaster-causing factors, disaster-forming environments, and disaster-bearing objects [39,40]. Based on the theory of natural disaster risk analysis and previous studies, social and economic factors affecting the occurrence of diseases and pests were considered in this paper. GDP per capita was selected as the impact indicator of disaster prevention and mitigation capacity, and the risk assessment system of diseases and pests was improved. The following is an illustration of its impact on disaster from the four aspects, and a representative and quantifiable impact factor is selected as the risk regionalization indicator.

(1) The danger of disaster-causing factors. Disaster-causing factors refer to all kinds of diseases and pests that cause the loss of food crops and are important risk assessment indicators. In this paper, the frequency of different degrees of corn thrips from 2013 to 2019 was selected (Figure 2) as the influence indicator of disaster-causing factors.

(2) The sensitivity of disaster-forming environments. Disaster-forming environments refer to the external environmental conditions that affect the growth and propagation of various diseases and pests. The influence indicators considered in this paper include the crop sown area, the meteorological condition, as well as the topography and landform, factors which lead to the occurrence of diseases and pests. The occurrence degree of diseases and pests is different due to the suitability of meteorological condition and topography and landform. This indicator has been included in the influence of disaster-causing factors and will not be repeated.

(3) The vulnerability of disaster-bearing objects. Disaster-bearing objects refer to the factor that affects crops when diseases and pests occur, and it is the entity of disaster-causing factors. In this study, the disaster-bearing object affected by corn thrips is the corn yield.

(4) Disaster prevention and mitigation capacity. The disaster prevention and mitigation capacity is also a socio-economic factor that restricts and influences the risk of diseases and pests. In this paper, GDP per capita was selected as the impact indicator of disaster prevention and mitigation capacity.

### 4.2. Indicator Weight Allocation

According to the steps of the cloud model, a hierarchical structure, including a target layer, criterion layer and indicator layer, was constructed. The criterion layer was divided into the danger of disaster-causing factors (C1), the sensitivity of disaster-forming environments (C2), the vulnerability of disaster-bearing objects (C3) and disaster prevention and mitigation capacity (C4). Each evaluation layer contained corresponding indicators, and the specific hierarchical structure is shown in Table 4.

**Table 4.** Hierarchical structural model.

| Target Layer | Criterion Layer | Indicator Layer |
|---|---|---|
| Risk assessment indicators for diseases and pests | The danger of disaster-causing factors (C1) | Frequency of severe occurrence (M1) Frequency of moderate occurrence (M2) Frequency of light occurrence (M3) |
| | The sensitivity of disaster-forming environments (C2) | Meteorological condition Topography and landform Crop sown area |
| | The vulnerability of disaster-bearing objects(C3) | Corn yield |
| | Disaster prevention and mitigation capacity (C4) | GDP per capita |

According to the cloud evaluation scale calculated using the golden section method, an expert group composed of 10 experts conducted a cloud evaluation on the importance of evaluation indicators. The cloud evaluation results of the criterion layer are shown in Figure 3.

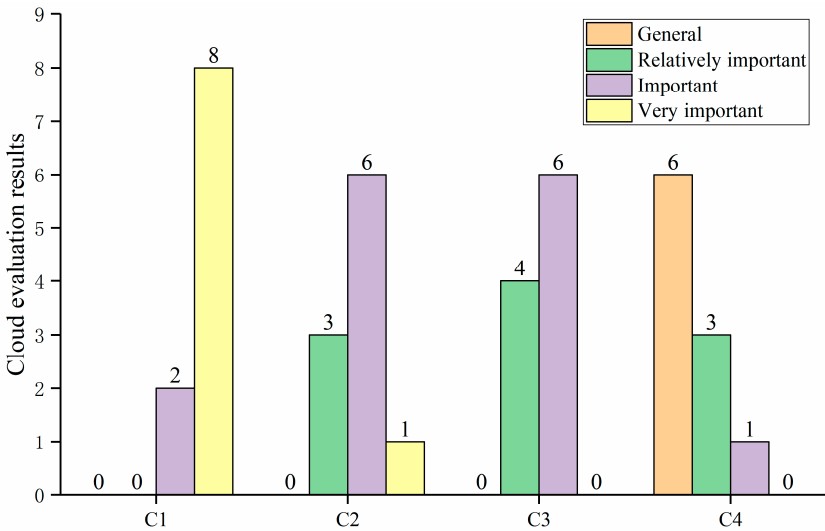

**Figure 3.** Cloud evaluation results.

Since the number of unimportant cloud evaluation results of each evaluation indicator is 0, this category is not displayed in the figure. Among them, the C1 criterion layer contains three indicators, which are evaluated in detail. The results are as follows: in the C1

indicator, the frequency of severe occurrence was considered very important by 9 experts and important by 1 expert; 7 experts considered frequency of moderate occurrence to be important, and 3 experts considered it to be relatively important. Frequency of light occurrence was considered relatively important by 5 experts and as general by other 5 experts. In C2, the weight of meteorological conditions and topography is assigned to 0, so there is no need for detailed evaluation.

The data of the evaluation indicators were brought into the model, the evaluation indicators were quantified through the cloud generator, and the relative importance between the evaluation indicators was calculated according to Equations (3)–(5). The quantitative results of C1 are shown in Table 5. The quantitative results of C2, C3 and C4 are shown in Table A1.

**Table 5.** Quantitative results of evaluation index C1.

| Parameters | V [1] | V [1] | V [1] | V [1] | V [1] | V [1] | V [1] | V [1] | I [2] | I [2] |
|---|---|---|---|---|---|---|---|---|---|---|
| Evaluation value ($x_i$) | 8.808 | 9.636 | 9.496 | 8.990 | 9.833 | 9.268 | 9.551 | 9.489 | 7.369 | 7.419 |
| Certainty degree ($\mu_i$) | 0.555 | 0.951 | 0.887 | 0.673 | 0.988 | 0.709 | 0.928 | 0.896 | 0.784 | 0.696 |
| Relative weight ($\omega_i$) | 0.069 | 0.118 | 0.110 | 0.083 | 0.122 | 0.088 | 0.115 | 0.111 | 0.097 | 0.086 |

[1] V: Very important; [2] I: Important.

Taking the calculation process of the C1 indicator as an example, as shown in Table 5, the cloud evaluation values of 10 experts were converted into cloud droplets ($x_i$, $\mu_i$) through the normal cloud model. We calculated the relative weight of the evaluation value which is very important according to Equation (3): $0.555/(0.555 + 0.951 + 0.887 + 0.673 + 0.988 + 0.709 + 0.928 + 0.896 + 0.784 + 0.696) = 0.069$. The importance of C1 was calculated to b 9.055211 according to Equation (4), and the relative importance was calculated by Equation (5) to be 0.35. The weights and comprehensive weights of risk assessment indicators were obtained according to relative importance, as shown in Table 6:

**Table 6.** Results of the calculation of weights for risk regionalization indicators for diseases and pests.

| Target Layer | Criterion Layer | | Indicator Layer | | Comprehensive Weights |
|---|---|---|---|---|---|
| | C1 | 0.35 | M1 | 0.475 | 0.16625 |
| | | | M2 | 0.319 | 0.11165 |
| | | | M3 | 0.206 | 0.0721 |
| Disaster risk assessment indicator system of corn thrips in Shandong Province | C2 | 0.263 | N1 | 0.263 | 0.263 |
| | | | N2 | 0 | 0 |
| | | | N3 | 0 | 0 |
| | C3 | 0.234 | / | | 0.234 |
| | C4 | 0.153 | / | | 0.153 |

*4.3. Disaster Risk Regionalization of Corn Thrips*

4.3.1. Risk Regionalization Results in Criterion Layer

(1)    Hazard regionalization

According to Equation (6), the hazard index of corn thrips was obtained, and the hazard index represents the hazard posed to each city by regionalization. Using the natural breakpoint classification method (Jenks), the hazard regionalization results of the corn thrip occurrence risk in the study area were obtained, as shown in Figure 4a. The occurrence frequency and harm degree of corn thrips in Heze city were the highest over the years assessed, followed by Jinan and Weihai City.

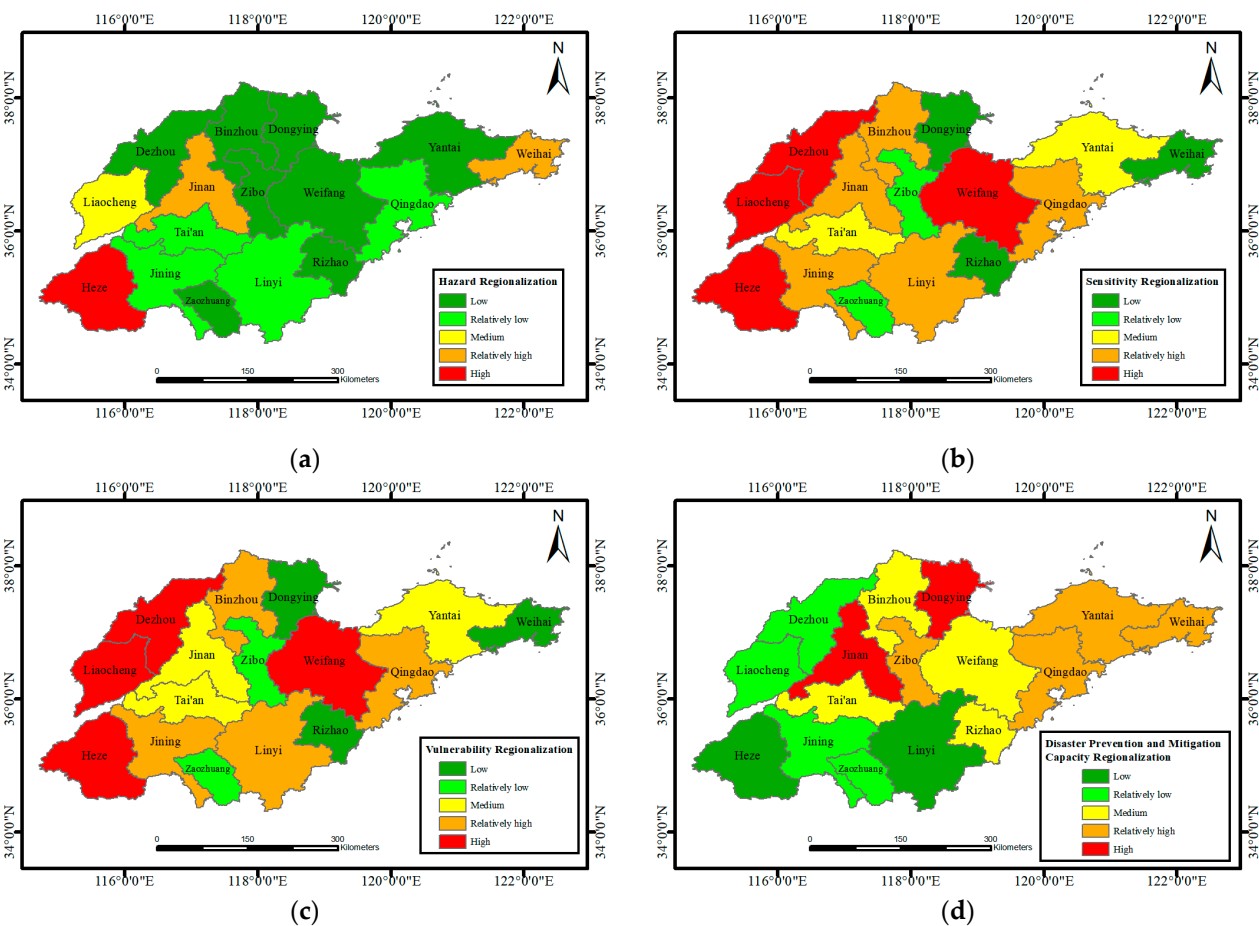

**Figure 4.** The disaster risk regionalization map of corn thrips in Shandong Province in criterion layer. (**a**) is the map of hazard regionalization, (**b**) is the map of sensitivity regionalization, (**c**) is the map of vulnerability regionalization, (**d**) is the map of disaster prevention and mitigation capacity regionalization.

(2) Sensitivity regionalization

According to Equation (7), the calculated value was the sensitivity index in various cities, and the sensitivity regionalization map was obtained according to the sensitivity index as shown in Figure 4b. High-sensitivity areas are located in Weifang, Dezhou, Heze and Liaocheng City. Weifang, Jining, Linyi, Binzhou and Qingdao City are relatively high-sensitivity areas. Generally, sensitivity is greater in the western region.

(3) Vulnerability regionalization

The higher the yield of corn thrips, the greater the loss, the stronger the vulnerability, and the greater the risk of corn thrips. The equation for the vulnerability index was the share of corn production in each municipality in the total production of Shandong Province, and the production values used were the average values for the years 2013 to 2019. The vulnerability regionalization map was obtained from the vulnerability index. The vulnerability regionalization is shown in Figure 4c. The west of Shandong Province and Weifang are the high corn-producing areas of our province and area also the high-risk areas for corn thrip occurrence. Weifang, Dezhou, Heze and Liaocheng are high-vulnerability areas. Jining, Qingdao, Binzhou and Linyi City are relatively high-vulnerability areas. The results are similar to those obtained for sensitivity regionalization.

(4) Disaster prevention and mitigation capacity regionalization

The capacity of disaster prevention and mitigation will also affect the occurrence of diseases and pests. Regions with high GDP per capita have strong capacity for disaster prevention and mitigation, so the probability of occurrence of diseases and pests will

decrease. The disaster prevention and mitigation capacity regionalization are shown in Figure 4d. The high-intensity area is Dongying and Jinan, followed by Qingdao, Weihai, Yantai and Zibo.

### 4.3.2. Comprehensive Risk Regionalization Result

Following the natural disaster risk indicator method, the comprehensive risk index within the study area was obtained. According to the comprehensive risk indicator, the comprehensive regionalization result is shown in Figure 5.

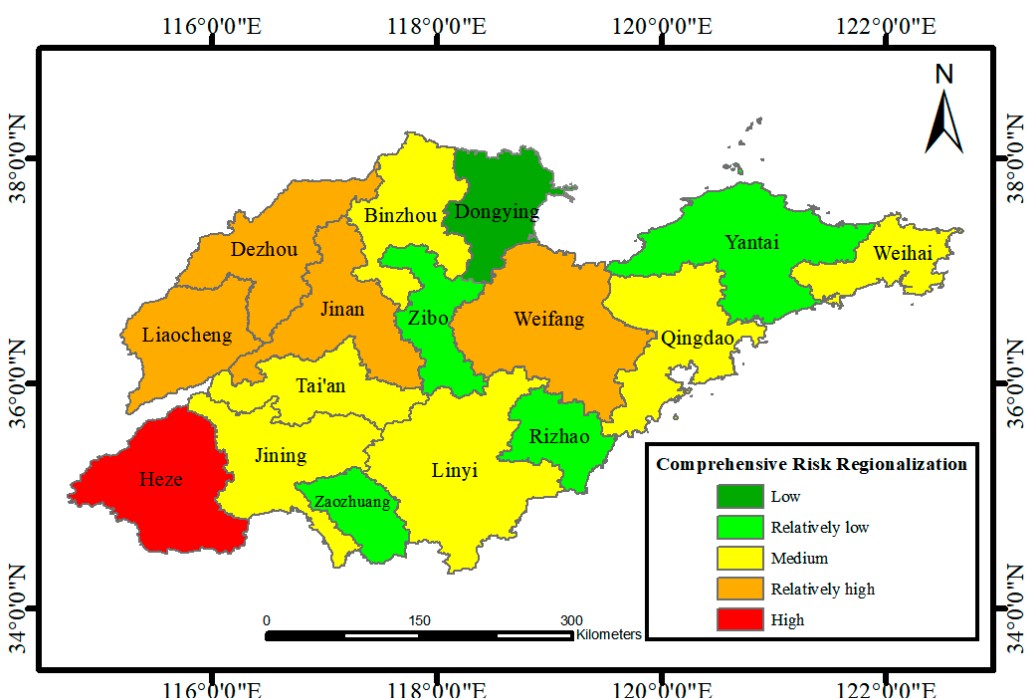

**Figure 5.** Comprehensive risk regionalization of corn thrips in Shandong Province (in target layer).

As can be seen from Figure 5, the comprehensive risk indicator is Heze, which has high yield, large sown area, high frequency of corn thrip occurrence and low disaster prevention and mitigation capacity, and is a high-risk area prone to corn thrips. Therefore, this area should be the key defense area against corn thrips. The smallest is Dongying, which has low yield, small sown area, low occurrence frequency of corn thrips, high GDP per capita, and strong disaster prevention and mitigation capacity. It is a low-risk area of corn thrip occurrence and is not susceptible to the harm caused by corn thrips. Secondly, Liaocheng, Dezhou, Jinan and Weifang City are relatively high-risk areas. Jining, Linyi, Qingdao, Binzhou, Tai'an and Weihai are medium-risk areas for the occurrence of corn thrips, with local outbreak risk. Other areas are relatively low-risk areas of thrip occurrence and less vulnerable to thrips. In general, the disaster risk of corn thrips in western Shandong Province is higher than that in other regions.

### 4.4. Disaster Risk Prediction in Corn Thrips

The disaster risk prediction results of the three models on the occurrence degree of corn thrips in 2018 and 2019 are shown in Figure 6. Where Figure 6a,b show the results predicted by multiple linear regression model in June and July, Figure 6c,d show the results predicted by GA-BP in June and July, and Figure 6e,f show the results predicted by GA-SVM in June and July.

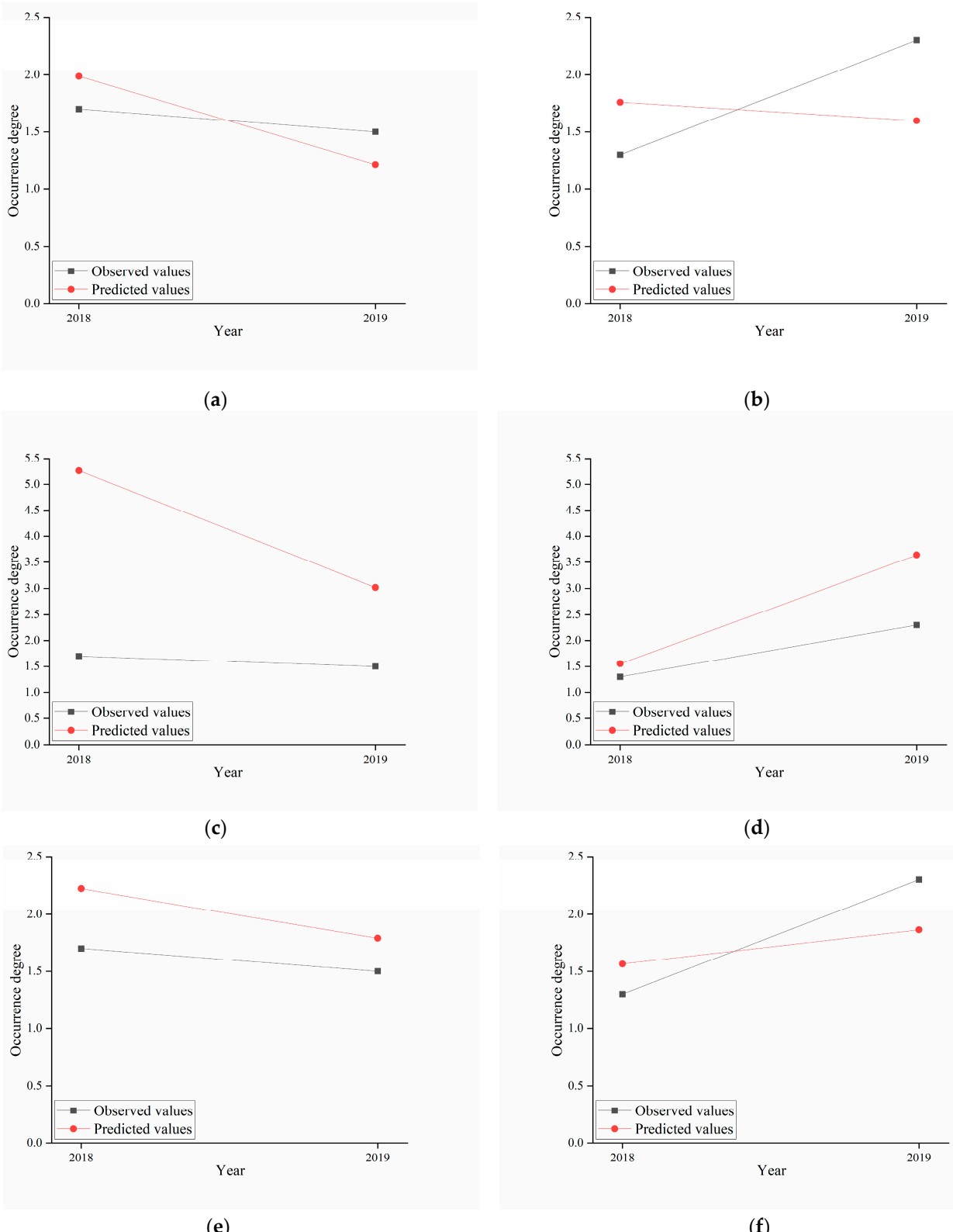

**Figure 6.** Disaster risk prediction results of occurrence degree of corn thrips in Shandong Province. (**a**,**b**) show the prediction results of multiple regression, (**c**,**d**) show the prediction results of genetic algorithm optimized back-propagation neural network, and (**e**,**f**) show the prediction results of genetic algorithm optimized support vector machine. (**a**,**c**,**e**) show prediction results in June, whereas (**b**,**d**,**f**) show prediction results in July.

As can be seen from Figure 6, the error of prediction results of multiple linear regression and GA-SVM models is within level 1, and the effect is good. The error of GA-BP model is large. The accuracy of the prediction results was verified by the real values. Specific prediction values and accuracy are shown in Table 7.

**Table 7.** Prediction values and accuracy. All values remain three decimal places.

| Methods | Date | True Value | Predicted Value | Prediction Error | Prediction Accuracy (%) | Average Accuracy (%) |
|---|---|---|---|---|---|---|
| Multiple linear regression | June 2018 | 1.7 | 1.987 | 0.287 | 85.556 | |
| | July 2018 | 1.3 | 1.759 | 0.459 | 73.906 | 72.971 |
| | June 2019 | 1.5 | 1.214 | −0.286 | 76.442 | |
| | July 2019 | 2.3 | 1.597 | 0.703 | 55.980 | |
| GA-BP | June 2018 | 1.7 | 5.265 | 3.565 | 32.289 | |
| | July 2018 | 1.3 | 1.553 | 0.253 | 83.709 | 57.239 |
| | June 2019 | 1.5 | 3.016 | 1.516 | 49.735 | |
| | July 2019 | 2.3 | 3.638 | 1.338 | 63.222 | |
| GA-SVM | June 2018 | 1.7 | 2.223 | 0.523 | 76.473 | |
| | July 2018 | 1.3 | 1.564 | 0.264 | 83.120 | 79.984 |
| | June 2019 | 1.5 | 1.790 | 0.290 | 83.799 | |
| | July 2019 | 2.3 | 1.863 | −0.437 | 76.543 | |

As can be seen from Table 7, the prediction effect of multiple linear regression on the occurrence degree of corn thrips is good, with an average accuracy of 72.971%. In June 2018, the prediction accuracy reached 85.556%. However, in July 2019, the accuracy was only 55.980%, indicating that the prediction effect is unstable. GA-BP prediction is poor, with an average accuracy of only 57.239%. The prediction effect was good only in July 2018, but very poor at other times. The average prediction accuracy of GA-SVM is 79.984%, with an accuracy exceeding 75% in all four periods and exceeding 80% in July 2018 and June 2019. The prediction accuracy is high and stable.

In general, the prediction effect of GA-SVM is the best. Compared with the multiple linear regression model and GA-BP, the accuracy is 7.013% and 22.745% higher, respectively. Among the three models, GA-SVM is the most suitable for predicting the disaster risk of corn thrips in Shandong Province, and the effect is accurate.

Through the correlation analysis of binary distance variables, it can be determine that the main meteorological factors affecting the occurrence of corn thrips are different in different months. In contrast to previous results [41], meteorological factors that are not usually considered, such as average water vapor pressure, maximum wind speed, and average air pressure, have become important factors affecting the occurrence of corn thrips. For example, the occurrence of corn thrips in June is related to average vapor pressure and maximum wind speed, whereas in July it is related to precipitation and average air pressure. Additionally, the temperature, which is often mentioned, does not show a particularly significant impact. Based on the backtesting of a multiple linear regression prediction model, the results showed that there is still a certain gap between the predicted results and the actual situation, and the stability of the model is poor. Compared with the prediction of the amount of second-generation corn thrips in Shandong Province using multiple linear regression by Chen et al. [41], the average prediction accuracy was 88.3%, with a maximum value higher than 98% and a minimum value of 54.89%, which is consistent with the results of this article. It is speculated that there are two main reasons for this: (1) the limitation of historical data, with a small amount of data and a narrow range, and the biological factors that affect prediction, including natural enemies and surrounding biological environments, not being considered. (2) The method currently used to establish the model being multiple linear regression, which means that non-linear correlation factors related to corn thrips are not considered. This indirectly affects the selection of predictive factors and the accuracy of the predictions.

The GA-BP model has a poor predictive fitting effect and low prediction accuracy, which is contrary to what was observed in the research results obtained by [42]. It was found that during the training process, repeated experiments were conducted several times, and the fitted values of the training samples were basically consistent with the true values, and the fitting effect was good. However, the results of the test samples differed greatly, which is called overfitting. The reason for overfitting is that the BP neural network is based on the principle of empirical risk minimization and requires a large amount of data samples to fully learn the "reasonable rules" between input and output data. However, the occurrence data of pests and diseases are typical small-sample data, with insufficient training samples and overtraining of the model. This results in the problems of overfitting and poor generalization ability. Therefore, the BP neural network is not suitable for building predictive models for small-sample data such as pests and diseases.

In this paper, the GA-SVM model has excellent prediction performance and strong stability. Unlike the BP neural network based on the principle of empirical risk minimization, the SVM is based on the principle of structural risk minimization and does not need to make any assumptions about the distribution properties of the data. It has strong generality and can perfectly solve the problems of small sample size and non-linearity in the occurrence system of pests, revealing the laws of pests and diseases occurrence.

## 5. Conclusions

In view of the problem that using a year as the time scale in previous studies could not accurately reflect the occurrence of corn thrips in each month, this paper used the month as the time scale and selected the evaluation indicators from four aspects to analyze the occurrence risk of corn thrips in Shandong Province by combining cloud model. Then, three models were used to predict the disaster risk of corn thrips in Shandong Province, and the accuracy of the prediction was quantitatively analyzed. The comprehensive risk regionalization results show that the high-risk area for the occurrence of corn thrips in Shandong Province is in Heze City; Liaocheng, Dezhou, Jinan and Weifang City are relatively high-risk areas. These cities should be regarded as the key defensive areas of corn thrips occurrence, and control should be carried out early in the growth process of corn. In addition, by comparing results with the true values, it can be seen the GA-SVM has the best effect in disaster risk prediction of corn thrips, with the average prediction accuracy of 79.984%, and that the model has good stability, meeting the basic requirements of prediction. Compared with the multiple linear regression model and GA-BP, the accuracy is 7.013% and 22.745% higher. The risk regionalization and prediction results of corn thrips disaster in Shandong Province can provide scientific basis for local corn thrips disaster management and theoretical support for corn thrips disaster regionalization and prediction. However, this study still needs further exploration and research in the following aspects. Due to the small amount of historical data, the influence of overwintering base, natural enemies, hosts, control measures, fertilization and other factors was not considered in this study. These areas should be added to the modeling process after qualitative and quantitative analysis of these factors through further experiments.

**Author Contributions:** Conceptualization, Y.Z. and M.J.; methodology, Y.Z.; software, M.J.; validation, Y.Z.; visualization, Y.Z. and Z.L.; investigation, Y.Z., Z.L. and F.J.; resources, F.J. and J.Y.; data curation, F.J.; formal analysis, Y.Z.; writing—original draft, Y.Z.; writing—review and editing, Z.L. and F.J.; supervision, J.Y.; project administration, M.J. and F.J.; funding acquisition, M.J. All authors have read and agreed to the published version of the manuscript.

**Funding:** This work was funded by Major Science and Technology Innovation Project in Shandong Province (2019JZZY020103).

**Data Availability Statement:** The data on the occurrence of corn thrips were obtained from Shandong Provincial Department of Agriculture and Rural Affairs, "Shandong Pesticide News", "Pesticide Market News", "Pesticide Express", and Qilu.com. Corn means of production, agricultural statistics and socio-economic statistics were obtained from "Shandong Statistical Yearbook". Meteorological data were obtained from China Weather Data Network.

**Acknowledgments:** The authors wish to thank Shandong Provincial Department of Agriculture and Rural Affairs for arranging corn thrip occurrence data.

**Conflicts of Interest:** The authors declare that they have no known competing financial interests or personal relationships that could have appeared to influence the work reported in this paper.

## Appendix A

**Table A1.** Quantitative results of evaluation index C2, C3 and C4.

| Evaluation Index | Parameters | V [1] | I [2] | I [2] | I [2] | I [2] | I [2] | I [2] | R [3] | R [3] | R [3] |
|---|---|---|---|---|---|---|---|---|---|---|---|
| C2 | Evaluation value ($x_i$) | 9.449 | 7.149 | 6.857 | 6.792 | 7.601 | 6.733 | 7.169 | 4.573 | 5.237 | 5.571 |
| | Certainty degree ($\mu_i$) | 0.858 | 0.949 | 0.997 | 0.981 | 0.508 | 0.965 | 0.934 | 0.624 | 0.875 | 0.415 |
| | Relative weight ($\omega_i$) | 0.106 | 0.117 | 0.123 | 0.121 | 0.063 | 0.119 | 0.115 | 0.077 | 0.108 | 0.051 |
| | / | V [1] | I [2] | I [2] | I [2] | I [2] | I [2] | R [3] | R [3] | R [3] | R [3] |
| C3 | Evaluation value ($x_i$) | 8.123 | 6.299 | 7.315 | 7.003 | 6.866 | 6.660 | 4.839 | 4.985 | 4.440 | 5.234 |
| | Certainty degree ($\mu_i$) | 0.116 | 0.564 | 0.800 | 0.985 | 0.997 | 0.945 | 0.897 | 0.999 | 0.445 | 0.858 |
| | Relative weight ($\omega_i$) | 0.015 | 0.074 | 0.105 | 0.129 | 0.131 | 0.124 | 0.118 | 0.131 | 0.058 | 0.113 |
| | / | I [2] | R [3] | R [3] | R [3] | G [4] | G [4] | G [4] | G [4] | G [4] | G [4] |
| C4 | Evaluation value ($x_i$) | 6.883 | 5.329 | 4.568 | 4.782 | 2.492 | 2.785 | 3.029 | 2.852 | 3.377 | 3.479 |
| | Certainty degree ($\mu_i$) | 0.999 | 0.670 | 0.554 | 0.810 | 0.674 | 0.909 | 0.996 | 0.938 | 0.917 | 0.797 |
| | Relative weight ($\omega_i$) | 0.121 | 0.081 | 0.067 | 0.098 | 0.082 | 0.110 | 0.120 | 0.114 | 0.111 | 0.096 |

[1] V: Very important; [2] I: Important; [3] R: Relatively important; [4] G: General.

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
