# Peer review of "Disaster Risk Regionalization and Prediction of Corn Thrips Combined with Cloud Model: A Case Study of Shandong Province, China"

_land, doi:10.3390/land12030709_

Round 1

Reviewer 1 Report

The manuscript reports the prediction of Corn Thrips prediction with cloud model. The topic is interesting and fits in the journal scope. I have following comments:

1.     Please use the full form in the abstract and avoid the abbreviations. 

2.     Please ensure that all the keywords are mentioned within the text.

3.     Choice of models being used in the manuscript needs justification. Why authors decided to use only the implemented models among many? Please provide a scientific justification within the introduction section of your choices and not the justification that literature uses these models. 

4.     I would suggest authors to include a section “experimental settings and evaluation measures” before the results sections where they should present the experimental protocols, model hyperparameters, training platform, training hardware, dataset split and evaluation measures used to assess the model performance.

5.     Authors need to subjectively discuss the performance of different models in terms of why they think a certain model performed better or worst. As of now, authors have just reported the results but not discussed. May be compare with literature and enhance the discussions to draw the conclusions out of the study.

A very well drafted last part of conclusion. However, I would suggest the authors to cut short the conclusions section and provide a concise single paragraph section. Also, listing so many limitations give an impression that the presented work is incomplete. Therefore, include only one or two main limitations in couple of lines. 

Author Response

尊敬的审稿人,

感谢您审阅我们题为“land-2262846”的手稿。这些富有成效的意见帮助我们大大改进了我们的稿件。我们对建议和意见进行了认真的修改。在附件中,我们详细说明了已修改的内容,这些更改不会影响论文的内容。

再次非常感谢您的意见和建议。

Reviewer 2 Report

Dear Authors,

I have carefully read your original manuscript.

In the attached file you will find some of my comments/suggestions which I ask you to respect in order to further improve the quality of the work.

Author Response

Dear reviewer,

  Thank you for your review of our manuscript entitled "land-2262846". These  fruitful comments help us improve our manuscript significantly. We have carefully revised these suggestions and comments. In the attachment, we detail these changes, and these changes will not affect the content of the paper.

  Once again, thank you very much for your comments and suggestions.
